# Identification and Spatial Distribution of Bioactive Compounds in Seeds *Vigna unguiculata* (L.) Walp. by Laser Microscopy and Tandem Mass Spectrometry

**DOI:** 10.3390/plants11162147

**Published:** 2022-08-18

**Authors:** Mayya P. Razgonova, Marina O. Burlyaeva, Yulia N. Zinchenko, Ekaterina A. Krylova, Olga A. Chunikhina, Natalia M. Ivanova, Alexander M. Zakharenko, Kirill S. Golokhvast

**Affiliations:** 1N.I. Vavilov All-Russian Institute of Plant Genetic Resources, 190000 Saint-Petersburg, Russia; 2Institute of Life Science and Biomedicine, Far Eastern Federal University, 690922 Vladivostok, Russia; 3Department of Botany, Saint-Petersburg State University, 199034 Saint-Petersburg, Russia; 4Siberian Federal Scientific Centre of Agrobiotechnology RAS, 633501 Krasnoobsk, Russia; 5Laboratory of Supercritical Fluid Research and Application in Agrobiotechnology, Tomsk State University, 634050 Tomsk, Russia

**Keywords:** *Vigna unguiculata*, tandem mass spectrometry, metabolites, local landrace, vegetable cultivar, grain cultivar, bioactive substances, seed, laser microscopy

## Abstract

The research presents a comparative metabolomic study of extracts of *Vigna unguiculata* seed samples from the collection of the N.I. Vavilov All-Russian Institute of Plant Genetic Resources. Analyzed samples related to different areas of use in agricultural production, belonging to different cultivar groups *sesquipedalis* (vegetable accessions) and *unguiculata* (grain accessions). Metabolome analysis was performed by liquid chromatography combined with ion trap mass spectrometry. Substances were localized in seeds using confocal and laser microscopy. As a result, 49 bioactive compounds were identified: flavonols, flavones, flavan-3-ols, anthocyanidin, phenolic acids, amino acids, monocarboxylic acids, aminobenzoic acids, fatty acids, lignans, carotenoid, sapogenins, steroids, etc. Steroidal alkaloids were identified in *V. unguiculata* seeds for the first time. The seed coat (palisade epidermis and parenchyma) is the richest in phenolic compounds. Comparison of seeds of varieties of different directions of use in terms of the number of bioactive substances identified revealed a significant superiority of vegetable accessions over grain ones in this indicator, 36 compounds were found in samples from cultivar group *sesquipedalis*, and 24 in *unguiculata*. The greatest variety of bioactive compounds was found in the vegetable accession k-640 from China.

## 1. Introduction

*Vigna unguiculata* (L.) Walp. is an important component of farming systems in many parts of the world. It is mainly grown on the continents of Africa, Asia, and South America. In recent years, information has appeared about the successful experience of its cultivation in the southern regions of Russia and Russian Far East [1,2]. *V. unguiculata* is a multipurpose vegetable crop and it is valued for its drought and heat tolerance. It is grown mainly for its seeds (cvg. *unguiculata* = ssp. *unguiculata*) or green vegetable pods (cvg. *sesquipedalis* = ssp. *sesquipedalis*). Food products prepared from *Vigna* are a source of many nutrients: proteins, amino acids, carbohydrates, minerals, fiber, vitamins, and other bioactive compounds [3,4,5,6,7].

This crop has a high level of polyphenols, some profiles of which are not commonly found in other legumes. The main polyphenols are phenolic acid derivatives (148–1176 µg/g) and flavonol glycosides (27–1060 µg/g). A number of varieties contain anthocyanins (875–3860 µg/g) and/or flavan-3-ols (2155–6297 µg/g). Monomers, mainly catechin-7-O-glucoside, predominate among the flavan-3-ols.

There are data on the content of bioactive peptides in *V. unguiculata*; although, their content varies depending on the variety. In addition, there is medical evidence showing significant anti-inflammatory effects and benefits of *V. unguiculata* polyphenols and peptides against cancer, diabetes, and cardiovascular diseases [8]. It holds great promise for wider use in modern food products due to nutritional properties that have a positive impact on health and a range of agronomic advantages over other legumes. The high content of polyphenols in the seeds, which are mainly concentrated in the seed coat, provides additional benefits for the use of phenolic extracts as nutraceutical and functional ingredients in food formulations [9].

The seeds of *V. unguiculata* are rich in bioactive compounds, and they can be used in the development of functional foods necessary for a healthy lifestyle. Such nutrition will serve as a medicine at the same time, because seeds of *V. unguiculata* have anti-inflammatory, immune-boosting, neuroprotective, anti-apoptotic, anti-cancer, antioxidant, anti-mutagenic, and cardioprotective properties [10].

At the present, analysis of the metabolomic composition of plants is applied for various purposes [11,12,13,14,15,16,17,18]. This approach is used to identify relationships between biochemical parameters and genetic characteristics of various crops, to solve various breeding problems, to characterize different groups of crops (variety types, subspecies, and species), to identify genotypes that do not differ morphologically and physiologically, etc. In recent years, interest has arisen in the application of metabolomic data in applied research aimed at solving problems in the food and pharmaceutical industries. Nutritional quality is becoming increasingly important for consumers and food manufacturers.

Comprehensive and diversified approaches are necessary to improve the quality of pods and seeds in different groups of *V. unguiculata* varieties, both vegetable and grain use. The data obtained from the analysis of the metabolome of seeds in the future can complement traditional and molecular genetic breeding methods that are aimed at creating new hybrids, donors of valuable traits, inbred lines, and varieties with high levels of bioactive compounds.

When searching for accessions with the highest nutritional value, and in order to create varieties with improved seed quality, it is important to study the content of bioactive substances, taking into account the specifics of their content in varieties with different directions of use. Microscopic methods, including confocal laser scanning microscopy, provide a unique opportunity to study the tissue localization of phytochemical compounds in plants. Comparison of HPLC data with fluorescent microscopy allows not only visualization of these compounds, but also the understanding of their function in plant organs. Currently, microscopic images are successfully used to clarify the morphological structure of cells and the structure of plant tissue [19], and to visualize the location of different groups of chemicals in plant organs [20,21,22]. However, there is limited information available on the presence and localization of bioactive compounds in *V. unguiculata* seeds.

The purpose of this research was a comparative metabolomic analysis of bioactive substances in *V. unguiculata* seeds derived from the collection of N.I. Vavilov All-Russian Institute of Plant Genetic Resources. Samples were grown in the field in Primorsky Krai (Russia) at the northern border of the crop area.

The objectives of the study were:-Analysis of bioactive compounds in seeds using high-performance liquid chromatography (HPLC) and tandem mass spectrometry (MS/MS) methods;-Visualization of the localization of phytochemical compounds in seed tissues using confocal laser microscopy;-Identification of differences in the content of bioactive compounds in seeds of vegetable and grain accessions (cultivar groups *sesquipedalis* and *unguiculata*).

## 2. Materials and Methods

### 2.1. Materials

The object of the research was the seeds of *V. unguiculata* from the group of varieties (cultivar groups) *sesquipedalis* and *unguiculata* harvested in 2020, grown at the Far East Experiment Station Branch of the Federal Research Center the N.I. Vavilov All-Russian Institute of Plant Genetic Resources (Table 1; Figure 1). Landraces were collected by N.I. Vavilov during the 1929 expedition to China (k-640, k-642) and obtained from the extract in 1921 from the USA (k-6) and in 1985 from Germany (k-1783); modern cultivar “Lyanchihe” (k-632341) was developed in Russia in the Primorsky Territory as a result of selection from samples of Chinese origin. Seeds (k-6) had cherry seed color; k-1783—beige; k-640, k-642, and k-632341—reddish-brown with dark strokes. Seeds for analysis were collected at the stage of industrial ripeness at the same time in 2020.

### 2.2. Chemicals and Reagents

HPLC-grade acetonitrile was purchased from Fisher Scientific (Southborough, UK), and MS-grade formic acid was from Sigma-Aldrich (Steinheim, Germany). Ultra-pure water was prepared from a SIEMENS ULTRA clear (SIEMENS water technologies, Munich, Germany), and all other chemicals were analytical grade.

### 2.3. Maceration

Fractional maceration technique was applied to obtain highly concentrated extracts. From 300 g of the sample, 4 g of *V. unguiculata* was randomly selected for maceration. The total amount of the extractant (ethyl alcohol of reagent grade) was divided into 3 parts, and the grains were consistently infused with the first, second, and third parts. The solid–solvent ratio was 1:20. The infusion of each part of the extractant lasted 7 days at room temperature.

### 2.4. Liquid Chromatography

HPLC was performed using Shimadzu LC-20 Prominence HPLC (Shimadzu, Kyoto, Japan), equipped with a UV-sensor and a Shodex ODP-40 4E reverse phase column to separate multicomponent mixtures. The gradient elution program was as follows: 0.01–4 min, 100% CH_3_CN; 4–60 min, 100–25% CH_3_CN; 60–75 min, 25–0% CH_3_CN; control washing 75–120 min 0% CH_3_CN. The entire HPLC analysis was performed using a UV–VIS detector SPD-20A (Shimadzu, Japan) at wavelengths of 230 and 330 nm, at 30 °C provided with column oven CTO-20A (Shimadzu, Japan) with an injection volume of 20 μL.

### 2.5. Mass Spectrometry

MS analysis was performed on an ion trap amaZon SL (BRUKER DALTONIKS, Bremen, Germany) equipped with an ESI source in negative ion mode. The optimized parameters were obtained as follows: ionization source temperature: 70 °C, gas flow: 4 L/min, nebulizer gas (atomizer): 7.3 psi, capillary voltage: 4500 V, end plate bend voltage: 1500 V, fragmentary: 280 V, collision energy: 60 eV. An ion trap was used in the scan range *m*/*z* 100–1.700 for MS and MS/MS. The mass spectra were recorded in negative and positive ion modes. The capture rate was one spectrum/s for MS and two spectrum/s for MS/MS. Data collection was controlled by Hystar Data Analysis 4.1 software (BRUKER DALTONIKS, Bremen, Germany). All experiments were repeated three times. A four-stage ion separation mode (MS/MS mode) was implemented. After a comparison of the *m*/*z* values, retention times, and fragmentation patterns with the MS/MS spectral data retrieved from the cited articles and after a database search (MS2T, MassBank, HMDB). The AmaZon SL ion trap is equipped with dedicated software to manage and interface it with 8 major HPLC system manufacturers. The Compass HyStar software (Version Bruker Compass HyStar 4.1 SR1 (4.1.28.0)) was used for synchronization with the Shimadzu chromatograph.

### 2.6. Optical Microscopy

The study of the structure of the *V. unguiculata* seed coat by light microscopy was carried out by performing sections of dry seeds.
Sections were prepared by hand with a safety razor from the middle part of half the seed in a direction perpendicular to the hilum. Photo fixation of sections and the study of the color of the seed coat were carried out in water, immediately after preparation of the sections.

For the confocal laser scanning microscopy, dry untreated *V. unguiculata* seeds were used. The transverse dissection was performed with an MS-2 sled microtome (Tochmedpribor, Kharkiv, Ukraine). The obtained sliced seeds were placed on microscopic cover glass through immersion oil to reduce light refraction by air gaps. The autofluorescence parameters were determined using confocal microscope (LSM 800, Carl Zeiss Microscopy GmbH, Berlin, Germany). The autofluorescence spectrum was chosen using lambda scan mode of the microscope, which allows to determine the emission maximum in a specific sample and obtain spectral acquisition. The specimen was excited by each laser separately and three main peaks of autofluorescence were revealed: excitation by a violet laser, 405 nm (solid state, diode, 5 mW) with the emission maximum of 400–475 nm (blue); excitation by a blue laser, 488 nm (solid state, diode, 10 mW) with the emission maxima in 500–545 nm (green) and 620–700 nm (red). The used power and detector gain for blue, green, and red channels were 5% and 750 V, 4.5% and 800 V, and 7% and 850 V, respectively. The objective Plan-Apochromat 63×/1.40 Oil DIC M27 with 63× magnification and the software ZEN 2.1 (Carl Zeiss Microscopy GmbH, Germany) were used for image acquisition and processing.

### 2.7. Statistical Data Processing

Statistical analysis included the compilation of binary matrices for each of the compounds identified in seeds of *V. unguiculata*, in which the “presence” (1) or “absence” (0) of the compound was noted in each of the studied samples. Based on the total matrix, a dendrogram was built, demonstrating the relationship between the studied samples. The method of unweighted pair-group cluster analysis with arithmetic averaging (UPGMA) using the TREECON program was used to construct the dendrogram. The cluster analysis was also carried out and a WPGMC (Median Clustering or Weighted Pair Group Method with Centroid Averaging) dendrogram was plotted in the Statistica 7 program, based on the data of the summary matrix.

In addition, based on the results of a comparative analysis of substances identified in *V. unguiculata* seeds, a Consensus tree was constructed using the Winclada-Nona program using the maximum parsimony criterion.

## 3. Results and Discussion

### 3.1. Tandem Mass Spectrometry

*V. unguiculata* extracts were analyzed using an ion trap coupled to high-performance liquid chromatography to better interpret the diversity of phytochemicals available. Primary analysis of the extracts showed a composition rich in bioactive substances. All experiments were repeated three times. A four-stage ion separation mode (MS/MS mode) was implemented. After a comparison of the *m*/*z* values, retention times, and fragmentation patterns with the MS/MS spectral data retrieved from the cited articles and after a database search (MS2T, MassBank, HMDB). Tentative identification showed the presence of 49 bioactive compounds detected by mass spectrometric analysis in *V. unguiculata* extracts. Forty-nine target analytes were successfully identified by comparing fragmentation patterns and retention times, most of which were polyphenols. Other compounds were identified by comparing their MS/MS data with the available literature. All identified compounds, along with molecular formulas, calculated and observed *m*/*z*, MS/MS data, and their comparative profile for *V. unguiculata*, are shown in Table 2.

Separately, it is worth noting the presence of steroidal alkaloids in all presented samples of *V.*
*unguiculata*, which were not previously noted by other authors. In addition, the presence of sapogenins A and B is also interesting, as they previously were identified in soybean (*Glycine* Willd.).

Figure 2 shows an example of decoding the spectrum of the steroidal alkaloid α-chaconine from an ion chromatogram obtained by tandem mass spectrometry. The [M + H]+ ion produces five product ions at *m*/*z* 706, *m*/*z* 673, *m*/*z* 560, *m*/*z* 437, and *m*/*z* 398 (Figure 2). A fragment ion at *m*/*z* 706 gives rise to two daughter ions at *m*/*z* 560 and *m*/*z* 398. A fragment ion at *m*/*z* 560 gives rise to five daughter ions at *m*/*z* 545, *m*/*z* 454, *m*/*z* 398, *m*/*z* 380, and *m*/*z* 213. This compound is identified in scientific articles as α-chaconine, for example, in *Solanum tuberosum* [72,77,78,80,81].

Table 3 shows the distribution of bioactive substances in accessions of *V. unguiculata*. Based on the tabular data, the sample of vegetable accession k-640 showed the greatest variety of bioactive compounds (29 compounds). In two other vegetable accessions (k-632341 and k-642), 19 and 13 compounds were found, respectively. In grain accessions (k-6 and k-1783), 18 and 7 compounds were identified, respectively.

Comparison of samples by the presence or absence of identified substances by different statistical methods did not reveal clear relationships with their origin and belonging to a certain group of varieties (cultivar groups) (Figure 3, Figure 4 and Figure 5). However, some samples differed from others in the presence of specific substances that were found only in them (Figure 6). Thus, only the grain accession “Clay” (k-6, USA) bred at the beginning of the last century contained the flavonol quercetin 3-*O*-glucoside, carotenoids *all-trans-*β-cryptoxanthin caprate, and *(all-E)*-violaxanthin myristate, tetrahydroxyflavan luteoliflavan-eriodictyol-*O*-hexoside. The steroidal alkaloid solanidadiene solatriose, omega-5 fatty acid myristoleic acid, and lignan dimethylmatairesinol were found in a local grain accession from Germany (k-1783). In addition, unlike other studied samples, only k-1783 lacked the anthocyanidin delphinidin 3-*O*-glucoside and sapogenin 3-rhamnose-galactose-glucuronic acid-soyasapogenol B. Vegetable accessions from China (k-640; k-642), and Primorsky Krai Russia (k-632341) also differed from each other in the content of some bioactive substances. Only k-642 had L-pyroglutamic acid and adenosine. Only k-632341 had aminobenzoic acid, protocatechuic acid, L-tryptophan, *(epi)*afzelechin and *(epi)*afzelechin-3-*O*-glucoside, omega-3 fatty acid linoleic acid. The lignan syringaresinol were not identified only in k-632341. The flavone acacetin, the steroidal alkaloid β-chaconine, and the phenolic acid *Trans*-salvianolic acid J were identified in k-640.

Despite the absence of a significant difference in the presence or absence of various bioactive compounds in seeds between grain and vegetable accessions, the samples from these groups differed in the number of identified compounds. Grain accessions had much fewer (24) bioactive substances than in vegetable samples (36) (Table 2 and Table 3). A comparison of accessions of different directions of use in terms of the number of identified classes of compounds also showed their smaller number in grain accessions (12) than in vegetable samples (23) (Figure 7).

It should be noted that indole-3-carboxylic acid, catechin, *(Epi)*afzelechin-4′-*O*-glucoside were found in all vegetable accessions and were not identified in grain accessions (Table 3).

Two compounds belonging to the group of anthocyanins were identified in vegetable accessions and one in grain accessions, and delphinidin 3-*O*-glucoside was not found only in k-1783 and k-640.

Five flavonoid compounds were identified: dihydrokaempferol in vegetable accession k-642 and grain accession k-6, quercetin 3-*O*-glucoside in k-6; myricetin in k-640; quercetin in k-632341; and dihydroquercetin in k-640 and in all grain accessions. Flavone acacetin has only been identified in vegetable cultivar k-640. A total of six flavan-3-ols were identified: catechin and *(epi)*afzelechin-4′-*O*-glucoside were found in all vegetable cultivars, *(epi)*afzelechin and *(epi)*afzelechin-3-*O*-glucoside were found only in vegetable cultivar k-632341, chinchonain Ia—in two vegetable accessions (k-642; k-632341) and one grain accession (k-6), *(epi)*catechin *O*-hexoside—in two vegetable accessions (k-632341; k-640). Tetrahydroxyflavan had only been identified in grain accession k-6 (luteoliflavan-eriodictyol-*O*-hexoside).

A class of phenolic acids has also been identified: trans-salvianolic acid J and coumaroyl quinic acid methyl ester were in k-640, salvianolic acid D was identified in k-632341 and k-640; protocatechuic acid was in k-632341, and a derivative of hydroxycinnamic acid (ferulic acid-*O*-hexoside) was found in k-640.

In addition, amino acids (L-tryptophan in k-632341), monocarboxylic acid (dihydroferulic acid in k-632341 and k-640), aminobenzoic acid (4-aminobenzoic acid in k-632341), carboxylic acid (indole-3-carboxylic acid in all vegetable accessions), non-proteinogenic L-α-amino acid (L-pyroglutamic acid in k-642), unsaturated monocarboxylic acid (9,10-dihydroxy-8-oxooctadec-12-enoic acid in k-6, and k-640), and trihydroxyoctadecadienoic acid (in k-640, k-632341, and k-6) were identified in *V. unguiculata* seeds.

From hydroperoxy fatty acids, hydroperoxy-octadecadienoic acids (k-640) were found; from long-chain fatty acids—nonacosanoic acid (k-6, k-640); from omega-3 fatty acids—linoleic acid (k-632341); from omega-5 fatty acid—myristoleic acid (k-1783); and from omega-hydroxy-long-chain fatty acid—hydroxy docosanoic acid (k-640).

In addition, three lignans (dimethylmatairesinol in k-1783, medioresinol in k-640, and syringaresinol in k-640), two carotenoids (*all-trans*-β-cryptoxanthin caprate, *(all-E)*-violaxanthin myristate, only in k-6), purine (Adenosine in k-642), and two sapogenins (3-Rhamnose-galactose-glucuronic acid-soyasapogenol B were identified in all accessions, except for k-1783; 6-deoxyhexose-hexoside-uronic acid-soyasapogenol A was found in k-642, k-640, and k-6.

Eight compounds from the group of steroidal alkaloids were also identified: solanidine (k-640, k-1783), β-chaconine (k-640), α-chaconine (in all samples except k-632341), α-solanine (k-642, k-640), solanidenol chacotriose (k-640; k-6), solanidadiene solatriose (k-1783), and solanidenediol chacotriose and leptinine II (k-640; k-6).

### 3.2. Confocal Laser Scanning Microscopy

Laser microscopy exploits the ability of the chemicals to fluoresce when excited by a laser and allows certain groups of chemical compounds in the plant tissues to be located. Our study allowed us to find out the spatial arrangement of phenolic compounds in the seed coat and cotyledons of *V. unguiculata*, based on the autofluorescence.

According to the literature data, the blue fluorescence in plants is mainly due to the presence of phenolic hydroxycinnamic acids [82]. The main fluorescent component is ferulic acid, but other hydroxycinnamic (*p*-coumaric, caffeic) acids can also contribute to it [83]. Moreover, lignin is a well-known source of blue fluorescence in plants. It has a wide emission range due to the presence of multiple fluorophore types within the molecule and can be observed when excited by UV and visible light [84]. Previous studies have shown that the lignin content of legume seed coat is low [85,86], and the cotyledons are poorly lignified [87]. Therefore, we suppose that most of the blue fluorescence in *V. unguiculata* seeds comes from hydroxycinnamic acids.

The blue-light-induced green autofluorescence in the range of 500–545 nm can be explained by the presence of flavins and flavonols (myricetin, quercetin, and kaempferol) and their derivatives [88,89,90]. The emission in the red spectrum mainly occurs due to the presence of anthocyanins and anthocyanidins [91,92,93].

The seed coat consists of cells of the palisade layer (palisade epidermis), hypoderma, and parenchyma. Sample k-6, with cherry-colored seeds, had a small number of flavonols and flavins in the cell walls of the palisade epidermis and in the cell cavities of this layer (Figure 8c). Anthocyanins and anthocyanidins accumulated mostly only in the parenchyma of the seed coat (Figure 8d). Phenolic acids were the most abundant in cotyledons, especially in their outer layer (Figure 8b).

According to confocal microscopy, the cavities and cell walls of the palisade epidermis of accession k-1783, which had a beige seed color, contained more flavonols and flavins than accession k-6 (Figure 9c). The same substances were found in the hypoderma and in several rows of parenchyma cells adjacent to it, as well as in the cells of the cotyledons. Anthocyanins and anthocyanidins were located in small inclusions in the cell cavities of the palisade epidermis and hypoderma and in the upper cells of the parenchymal layer (Figure 9d). Quite significant differences between the two grain samples can be noted in the location of phenolic acids. Phenolic acids were almost exclusively in the cotyledons in accession k-6 (Figure 8b), and the same acids were in the cotyledons and in the cells of the parenchyma of the seed coat in accession k-1783 (Figure 9b).

Despite the light beige color of the seeds, the k-1783 accession had much more bioactive substances in the seed coat than the k-6 sample with dark cherry seeds.

Photographs showed a more intense coloration of the palisade epidermis and underlying layers of the seed coat in vegetable accessions (k-640, k-642, and k-632341), in contrast to grain accessions (Figure 10, Figure 11 and Figure 12). The seeds of these specimens were reddish-brown in color and had longitudinal dark streaks running parallel to the hilum. Anthocyanins and anthocyanidins were present in the palisade epidermis, in the hypoderma, and in the cells of the parenchyma adjacent to the hypodermis. Among vegetable samples, k-640 had a less bright red color, and k-632341 had a stronger red color (characterized by dark cherry pods at the stage of technical ripeness). The green color, indicating the presence of flavonols and flavins, was the most intense for k-642, less for k-640, and k-632341. These compounds were located in all vegetable accessions both in the palisade epidermis, hypoderma, parenchyma, and in the cotyledons (Figure 10c, Figure 11c, and Figure 12c).

Phenolic acids in all vegetable samples were concentrated to a greater extent in the cotyledons, and to a lesser extent in the seed coat (Figure 10b, Figure 11b and Figure 12b). It should be noted that in k-642 and k-632341, the hypoderma and parenchyma of the seed coat were more strongly colored blue than in k-640. Based on the photographs, we can conclude that the highest content of the studied substances was found in accession k-642.

A large number of researchers have shown the important role of the seed coat in supplying the embryo with nutrients during development [94,95]. It plays a significant role in the regulation of seed dormancy and germination, and it is a rich source of many valuable substances. It contains a wide range of compounds: flavonoids, proteins, peptides, amino acids, alkaloids, terpenoids, steroids, etc. [96]. Many components of the seed coat play an important role in seed protection. As a result of our study of seeds using confocal microscopy, it was found that in most samples, the largest number of bioactive substances was in the palisade epidermis of the seed coat, and fewer compounds were found in the hypoderma, parenchyma, and cotyledons. Moreover, phenolic acids were localized mainly in the cotyledons, less in the parenchyma of the seed coat. Flavonols and flavins were located mainly in the palisade epidermis, and less in the cells of the hypoderma, parenchyma of the seed coat, and cotyledons. Anthocyanins and anthocyanidins in vegetable samples (k-640, k-642, k-632341) were present not only in the palisade epidermis, but also in parenchyma cells. Anthocyanins were found mainly in parenchyma cells and hypodermis in grain accessions (k-6, k-1783)

Polyphenolic compounds, including phenolic acids and their derivatives, tannins, and flavonoids, represent the largest group of natural plant nutrients. They determine the color of fruits and seeds and play an important role in disease resistance [97,98]. Most of the phenolic compounds found in legume seeds are also located in the seed coat. In soybean and common bean, the concentration of phenolic compounds such as flavonoids and anthocyanins correlate with seed coat color [99,100]. In our study, according to the results of tandem mass spectrometry, anthocyanidins, which determined the color of cherry and red-brown seeds, were identified in *V. unguiculata* seeds. The darkest seeds (cherry) of accession k-6 had the smallest number of different polyphenols, compared with the seeds of vegetable accessions (k-640, k-642, and k-632341), colored red-brown. It can be assumed that the color of the seed coat of k-6 also depends on the carotenoid (*(all-trans)*-β-cryptoxanthin caprate), which was found only in its seeds.

A comparison of the results obtained using confocal laser microscopy with data on the content of bioactive substances in seeds identified using tandem mass spectrometry provides additional information. Laser microscopy makes it possible to visualize the comparative concentration of substances in plant tissues, which is an additional characteristic in the study of bioactive substances in samples. If, according to the data of tandem mass spectrometry, k-640 had the greatest variety of identified substances, then k-642 was distinguished by the concentration of anthocyanidins and flavonols in the cells of the seed coat.

## 4. Conclusions

The seeds of both vegetable and grain accessions of *V. unguiculata* are rich in bioactive compounds: phenols, polyphenols, flavonols, flavones, anthocyanins, amino acids, carotenoids, omega-3 and 5 fatty acids, sapogenins, steroids, etc. We identified 49 bioactive substances, most of which belonged to the class of polyphenols. For the first time, steroidal alkaloids were found in *V. unguiculata* seeds, and they were present in all studied samples. Most of the bioactive substances were localized in the palisade epidermis; the smaller part was in the hypoderma and parenchyma of the seed coat and cotyledons. Seeds of vegetable accessions differed from seeds of grain accessions in a large number of bioactive substances, 36 and 24, respectively. Comparison of accessions of different directions of use in terms of the number of classes of compounds also showed their smaller number in grain accessions (13) than in vegetable accessions (22).

Given these differences, it is the most effective to include seeds from accessions of different uses in different diets. Vegetable accessions differed in the content of a larger number of compounds related to flavan-3-ol, anthocyanidin, lignan, phenolic acid, etc., only carotenoid was encountered in grain varieties. Further research involving a larger number of samples will provide new data on the regularities of the content of substances in accessions from different use groups and apply them in the development of dietary recommendations, as well as in the selection of varieties with improved seed quality.

## Figures and Tables

**Figure 1 plants-11-02147-f001:**
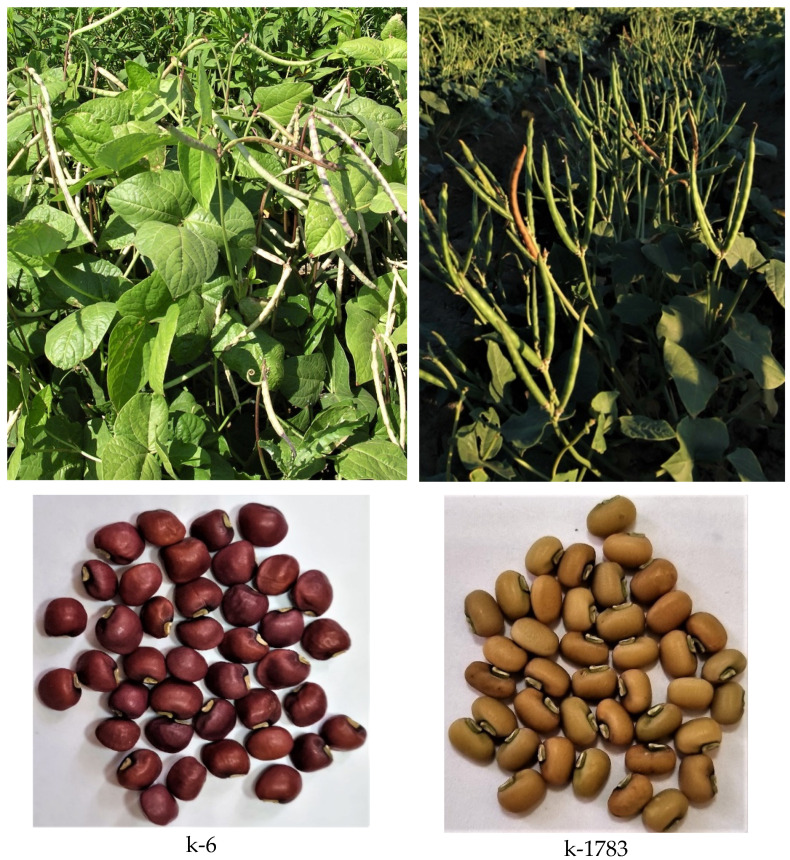
Samples of *V. unguiculata* grown at the Far East Experiment Station Branch of the Federal Research Center the N.I. Vavilov All-Russian Institute of Plant Genetic Resources. Appearance of the plants and seeds.

**Figure 2 plants-11-02147-f002:**
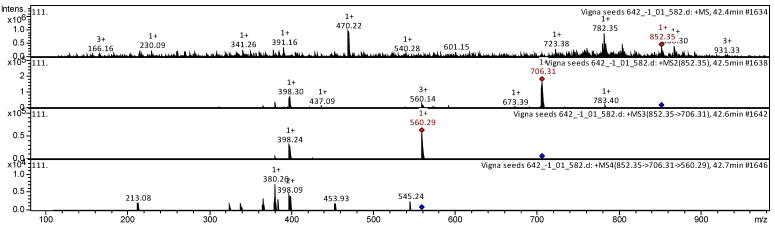
Mass spectrum of the steroidal alkaloid α-chaconine from *V. unguiculata* extract, at *m*/*z* 852.35.

**Figure 3 plants-11-02147-f003:**
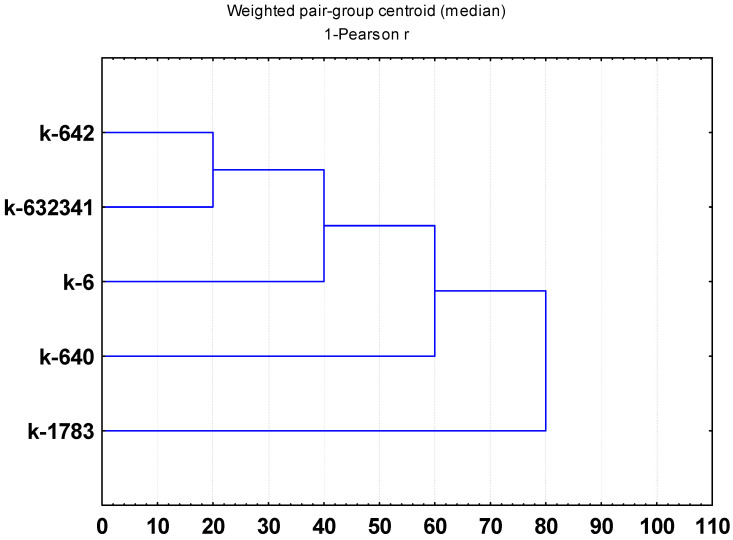
Dendrogram WPGMC (Median Clustering or Weighted Pair Group Method with Centroid Averaging), plotting on the basis of a comparative analysis of substances identified in *V. unguiculata* seeds.

**Figure 4 plants-11-02147-f004:**
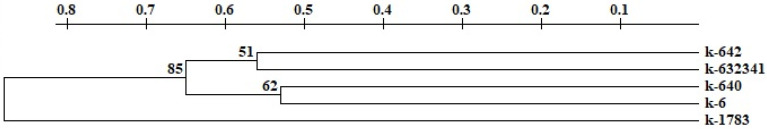
Dendrogram UPGMA (Unweighted Group Average or Unweighted Pair Group Method with Arithmetic Averaging), constructed on the basis of a comparative analysis of substances identified in *V. unguiculata* seeds.

**Figure 5 plants-11-02147-f005:**
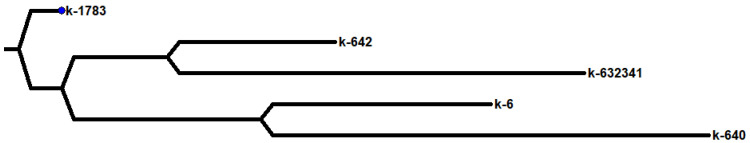
Consensus tree built using the criterion of maximum parsimony based on the results of a comparative analysis of substances identified in *V. unguiculata* seeds (Ci = 77, consistency index—the proportion of homoplasia in the total number of changes in traits, L = 63, Ri = 33 retention index—the number of synapomorphies determined by the data).

**Figure 6 plants-11-02147-f006:**
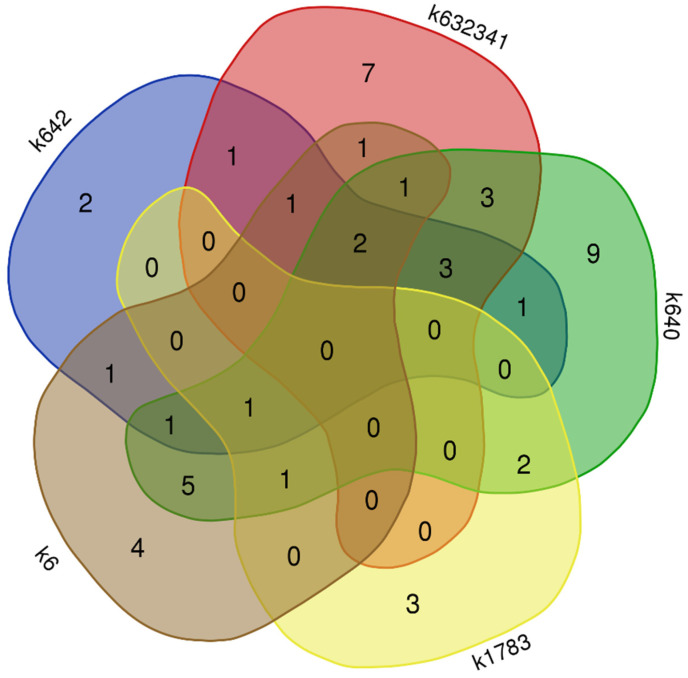
Venn diagram of bioactive compounds between *V.*
*unguiculata* accessions.

**Figure 7 plants-11-02147-f007:**
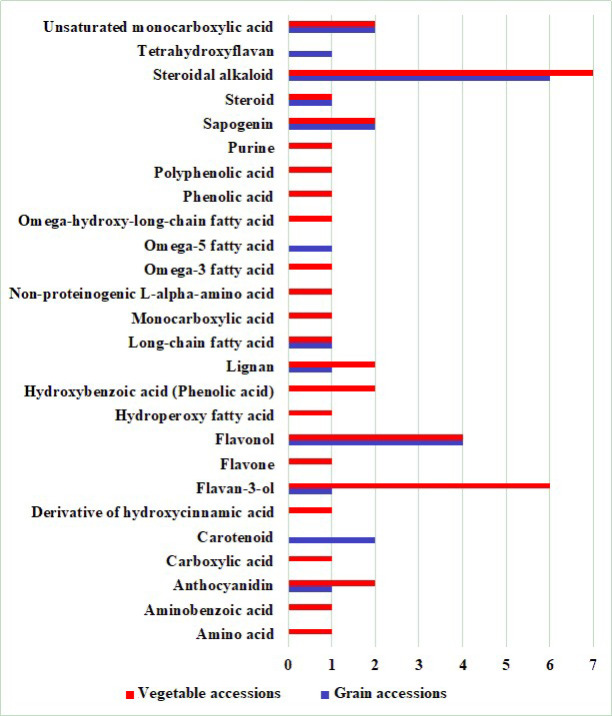
The content of bioactive substances in the seeds of grain and vegetable accessions. 1–7—number of identified substances.

**Figure 8 plants-11-02147-f008:**
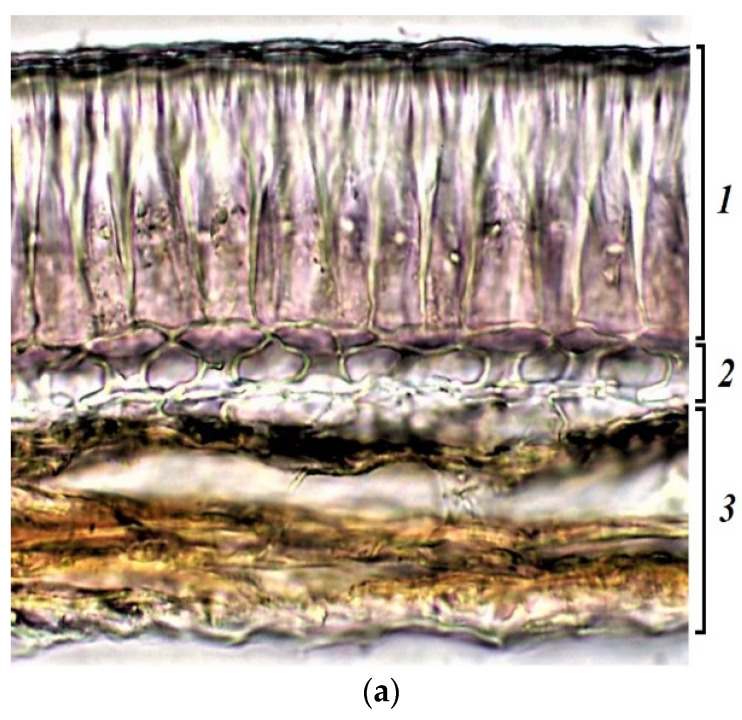
*V.**unguiculata* accession k-6: (**a**)—seed coat structure, light microscopy (1—palisade layer, 2—hypoderma, 3—parenchyma); (**b**–**e**) transverse section of the seed, confocal microscopy, (**b**)—excitation 405 nm with the emission in 400–475 nm (blue), (**c**)—excitation 488 nm with the emission in 500–545 nm (green), (**d**)—excitation 488 nm with the emission in 620–700 nm (red), (**e**)—merged.

**Figure 9 plants-11-02147-f009:**
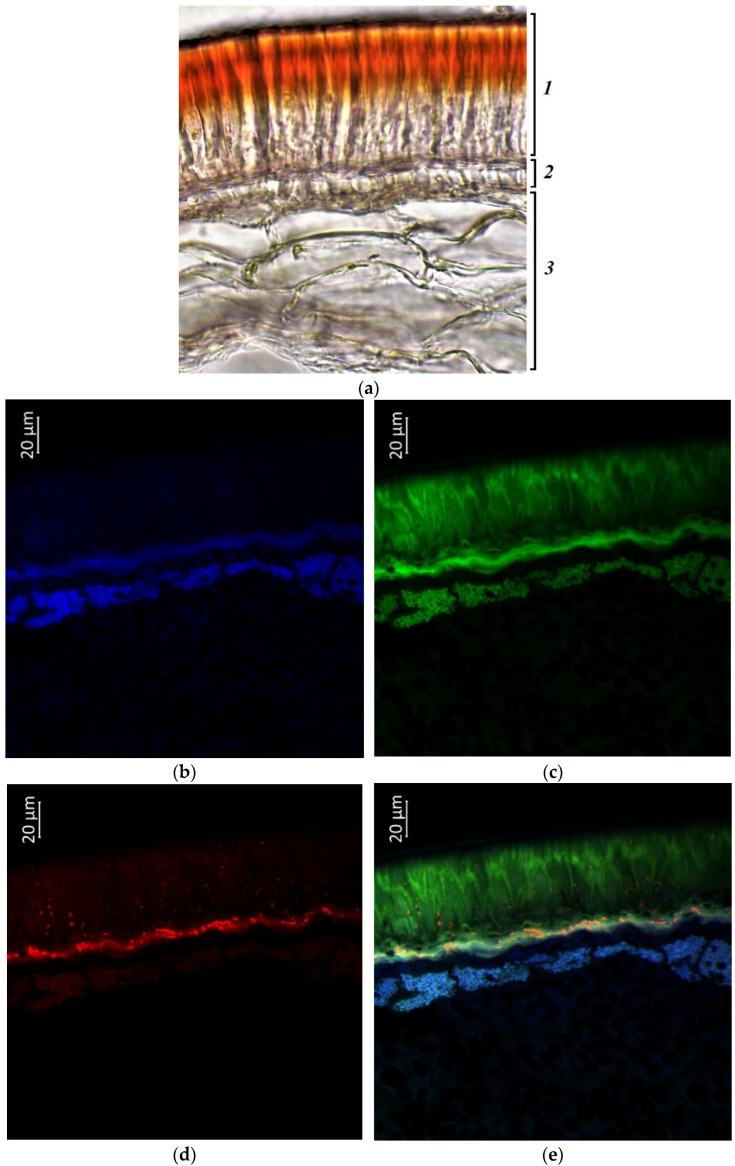
*V. unguiculata* accession k-1783: (**a**)—seed coat structure, light microscopy (1—palisade layer, 2—hypoderma, 3—parenchyma); (**b**–**e**)—transverse section of the seed, confocal microscopy, (**b**)—excitation 405 nm with the emission in 400–475 nm (blue), (**c**)—excitation 488 nm with the emission in 500–545 nm (green), (**d**)—excitation 488 nm with the emission in 620–700 nm (red), (**e**)—merged.

**Figure 10 plants-11-02147-f010:**
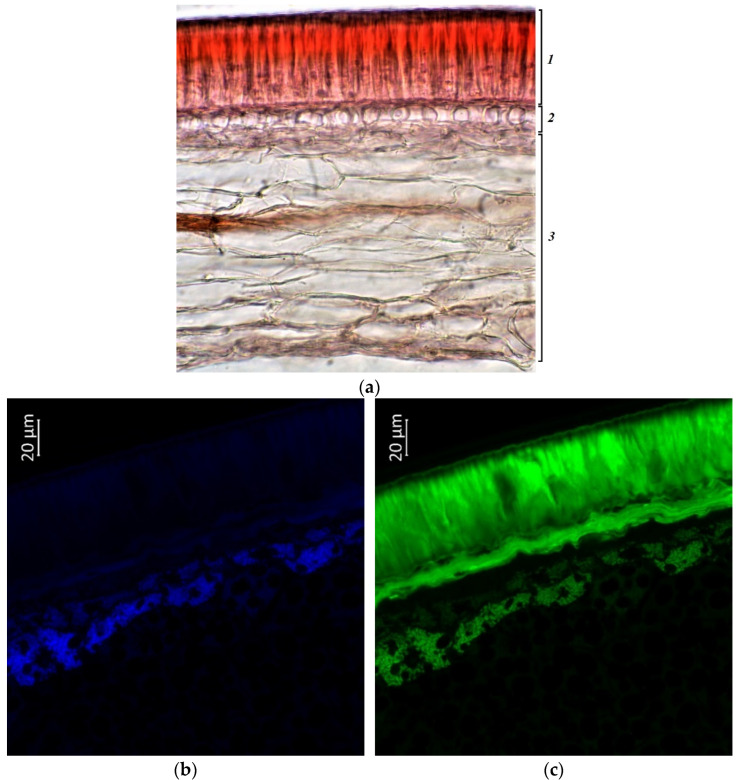
*V.**unguiculata* accession k-640: (**a**)—seed coat structure, light microscopy (1—palisade layer, 2—hypoderma, 3—parenchyma); (**b**–**e**)—transverse section of the seed, confocal microscopy, (**b**)—excitation 405 nm with the emission in 400–475 nm (blue), (**c**)—excitation 488 nm with the emission in 500–545 nm (green), (**d**)—excitation 488 nm with the emission in 620–700 nm (red), (**e**)—merged.

**Figure 11 plants-11-02147-f011:**
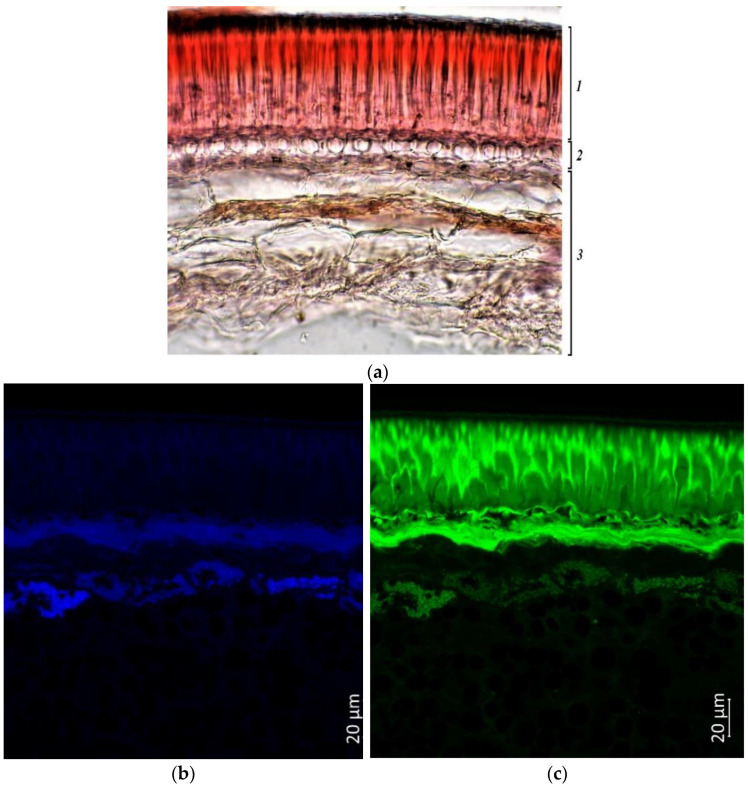
*V*. *unguiculata* accession k-642: (**a**)—seed coat structure, light microscopy (1—palisade layer, 2—hypoderma, 3—parenchyma); (**b**–**e**) transverse section of the seed, confocal microscopy, (**b**)—excitation 405 nm with the emission in 400–475 nm (blue), (**c**)—excitation 488 nm with the emission in 500–545 nm (green), (**d**)—excitation 488 nm with the emission in 620–700 nm (red), (**e**)—merged.

**Figure 12 plants-11-02147-f012:**
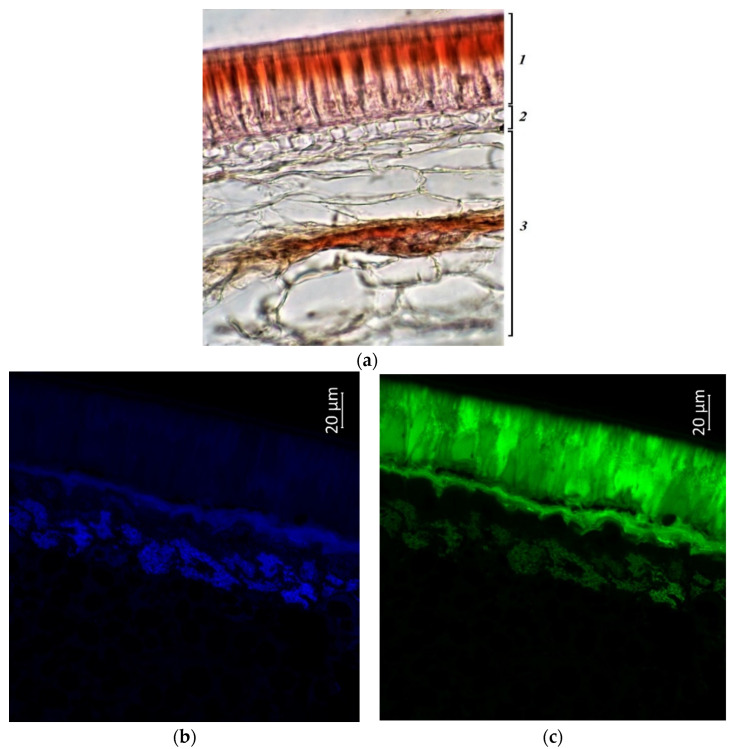
*V.**unguiculata* accession k-632341: (**a**)—seed coat structure, light microscopy (1—palisade layer, 2—hypoderma, 3—parenchyma); (**b**–**e**)—transverse section of the seed, confocal microscopy, (**b**)—excitation 405 nm with the emission in 400–475 nm (blue), (**c**)—excitation 488 nm with the emission in 500–545 nm (green), (**d**)—excitation 488 nm with the emission in 620–700 nm (red), (**e**)—merged.

**Table 1 plants-11-02147-t001:** *V. unguiculata* seed material samples.

No	VIR Catalogue Number	Name of Accessions	Country of Origin	Acqdate	Cultivar Groups
1	k-6	Cultivar “Clay”	USA	1921	*unguiculata*
2	k-640	Landrace	China	1929	*sesquipedalis*
3	k-642	Landrace	China	1929	*sesquipedalis*
4	k-1783	Landrace	Germany	1985	*unguiculata*
5	k-632341	Cultivar “Lyanchihe”	Far East, Russia	2018	*sesquipedalis*

**Table 2 plants-11-02147-t002:** Compounds identified from extracts of *V. unguiculata* under positive and negative ionization modes by tandem mass spectrometry.

No	VIR Catalogue Number	Class of Compounds	Identified Compounds	Formula	Mass	Molecular Ion [M-H]-	Molecular Ion [M+H]+	2 Fragmentation MS/MS	3 Fragmentation MS/MS	4 Fragmentation MS/MS	References
	**Polyphenols**										
1	k-6(583); k-642 (582)	Flavonol	Dihydrokaempferol (Aromadendrin; Katuranin)	C_15_H_12_O_6_	288.25	287		151; 269			*Solanium tuberosum* [23]; *F. glaucescens* [24]; *Camellia kucha* [25]; *Echinops* [26]
2	k-6(583); k-632341 (579)	Flavonol	Quercetin	C_15_H_10_O_7_	302.23	301		179; 273	151;		Potato leaves [27]; *Vigna sinensis* [28]; *Vaccinium macrocarpon* [29]; Propolis [30]
3	k-6(583); k-1783 (585); k-640 (589); k-640 (590)	Flavonol	Dihydroquercetin (Taxifolin; Taxifoliol)	C_15_H_12_O_7_	304.25	303		285; 177	241		*Dracocephalum palmatum* [31]; *Vitis amurensis* [32]; *Rhodiola rosea* [33]
4	k-640 (590)	Flavonol	Myricetin	C_15_H_10_O_8_	318.23	317		273	260; 251		*Vaccinium macrocarpon* [29]; *F. glaucescens* [24]; millet grains [34]; *Sanguisorba officinalis* [35]
5	k-6(584)	Flavonol	Quercetin 3-*O*-glucoside (Isoquercitrin; Hirsutrin)	C_21_H_20_O_12_	464.37		303	256; 165	229		Potato [23]; *Vigna sinensis* [28]; Andean blueberry [36]; *Lonicera Henryl* [37]
6	k-640 (590)	Flavone	Acacetin (Linarigenin; Buddleoflavonol)	C_16_H_12_O_5_	284.26		285	257; 239; 177	248; 237; 216; 173		*Mentha* [38]; *Dracocephalum palmatum* [39]; *Wissadula periplocifolia* [40]
7	k-6(583)	Tetrahydroxyflavane	Luteoliflavan-eriodictyol-O-hexoside	C_36_H_34_O_16_	722.64		723	587; 555; 499	543; 516; 499	499	*C. edulis* [24]
8	k-632341 (579)	Flavan-3-ol	Epiafzelechin ((epi)Afzelechin)	C_15_H_14_O_5_	274.26		275	195; 149	167	150	*Cassia granidis* [41]; *Cassia abbreviata* [42,43]; *A. cordifolia; F. glaucescens; F. herrerae* [24]
9	k-632341 (579); k-632341 (580); k-640 (590); k-642 (582)	Flavan-3-ol	Catechin (D-Catechol)	C_15_H_14_O_6_	290.26	289		245; 205	201	175	*Eucalyptus* [44]; *Vaccinium macrocarpon* [45]; *C. edulis* [24]; *Vigna inguiculata* [46]; *Triticum* [47]
10	k-632341 (579); k-640 (590); k-642 (582)	Flavan-3-ol	(Epi)afzelechin-4′-*O*-glucoside	C_21_H_24_O_10_	436.41	435		299; 191; 161	151; 117		*Vigna unguiculata* [46]
11	k-632341 (580)	Flavan-3-ol	(Epi)afzelechin-3-*O*-glucoside	C_21_H_24_O_10_	436.41	435		313; 299; 273			*Vigna unguiculata* [46]; *Cassia abbreviata* [42]
12	k-6(583); k-6(584); k-632341 (579); k-642 (582)	Flavan-3-ol	Chinchonain Ia	C_24_H_20_O_9_	452.41	451		289	245	203	Andean blueberry [36]
13	k-632341 (579); k-632341 (580); k-640 (590)	Flavan-3-ol	(epi)Catechin *O*-hexoside	C_21_H_24_O_11_	452.41	451		289; 269; 245	245; 231	227	Andean blueberry [36]
14	k-6(583); k-6(584); k-632341 (579); k-632341 (580); k-640 (589); k-640 (590); k-642 (582)	Anthocyanidin	Delphinidin 3-*O*-glucoside	C_21_H_21_O_12+_	465.39	463		300	151; 271	169	Rapeseed petals [48]; *Vigna sinensis* [28]; *Berberis ilicifolia; Berberis empetrifolia; Ribes maellanicum; Ribes cucullatum; Myrteola nummalaria* [49]; *Vigna unguiculata* [50]
15	k-632341 (579); k-632341 (580); k-642 (582)	Anthocyanidin	Delphinidin-3,5-*O*-diglucoside	C_27_H_30_O_17_	626.52		626	303; 465	257; 165	229; 157	*Vitis labrusca* [51]; *Solanium nigrum* [52]; Muscadine pomace [53]
16	k-1783 (585)	Lignan	Dimethylmatairesinol(Arctigenin Methyl Ether)	C_22_H_26_O_6_	386.44		387	205			Lignans [54]
17	k-640 (590)	Lignan	Medioresinol	C_21_H_24_O_7_	388.41	387		207; 225; 179			Lignans [54]; *Punica granatum* [55]; *Bituminaria* [56]
18	k-632341 (579)	Lignan	Syringaresinol	C_22_H_26_O_8_	418.44		419	326; 248; 151	298; 254; 218; 174	251; 182; 145	*Triticum aestivum* L. [47]; *Lignans* [54]; *Punica granatum* [55]; *Magnolia thailandica* [57]
19	k-632341 (579)	Hydroxybenzoic acid (Phenolic acid)	Protocatechuic acid	C_7_H_6_O_4_	154.12		155	126			*Vigna unguiculata* [6]; *Eucalyptus* [44]; *Eucalyptus Globulus* [58]; *Vaccinium macrocarpon* [45]; *Lonicera japonica* [59]
20	k-640 (590)	Polyphenolic acid	Coumaroyl quinic acid methyl ester	C_17_H_20_O_8_	352.34	351		285; 267; 243	242; 200		*F. glaucescens* [24]
21	k-640 (590)	Derivative of hydroxycinnamic acid	Ferulic acid-*O*-hexoside	C_16_H_20_O_9_	356.32	355		191;209; 174	173		*A. cordifolia* [24]; millet grains [34]; Rapeseed petals [48]; beer [60]; strawberry [61]
22	k-632341 (579); k-640 (589)	Hydroxybenzoic acid	Salvianolic acid D	C_20_H_18_O_10_	418.35	417		373	347	303	*Salvia miltiorrhiza* [62]; *Lonicera caerulea* [63]
23	k-640 (590)	Phenolic acid	*Trans*-salvianolic acid J	C_27_H_22_O_12_	538.46		539	493; 479; 357	420		*Mentha* [38]
	**Others**										
24	k-642 (582)	Non-proteinogenic L-α-amino acid	L-Pyroglutamic acid (Pidolic acid; 5-Oxo-L-Proline)	C_5_H_7_NO_3_	129.11		130	112			Potato leaves [27]
25	k-632341 (580)	Aminobenzoic acid	4-Aminobenzoic acid (*p*-aminobenzoic acid)	C_7_H_7_NO_2_	137.14		138	119			*Solanum tuberosum* [23]
26	k-632341 (579) k-632341 (580) k-640 (589); k-640 (590) k-642 (582)	Carboxylic acid	Indole-3-carboxylic acid	C_10_H_9_NO_2_	175.18		176	159; 130			Beer [60]
27	k-632341 (579); k-632341 (580); k-640 (590)	Monocarboxylic acid	Dihydroferulic acid	C_10_H_12_O_4_	196.2	195		159; 129			*A. cordifolia* [24]; Coffee [64]
28	k-632341 (579); k-632341 (580)	Amino acid	L-Tryptophan (Tryptophan; (S)-Tryptophan)	C_11_H_12_N_2_O_2_	204.23		205	188	146; 144	118	*Camellia kucha* [25]; *Vigna inguiculata* [6,46]; Rapeseed petals [48]; *Perilla frutescens* [65]
29	k-1783 (585)	Omega-5 fatty acid	Myristoleic acid (Cis-9-Tetradecanoic acid)	C_14_H_26_O_2_	226.36		227	209	139	122	*F. glaucescens* [24]
30	k-642 (582)	Purine	Adenosine	C_10_H_13_N_5_O_4_	267.24		268	136			*Lonicera japonica* [59]
31	k-632341 (579)	Omega-3 fatty acid	Linoleic acid (Linolic acid; Telfairic acid)	C_18_H_32_O_2_	280.45	279		261; 205	205		Salviae [66]; *Angelicae sinensis Radix* [67]; *Pinus sylvestris* [68]
32	k-640 (590)	Hydroperoxy fatty acid	Hydroperoxy-octadecadienoic acid	C_18_H_32_O_4_	312.44	311		183; 309			Potato [23]
33	k-6(583); k-640 (589)	Unsaturated monocarboxylic acid	9,10-Dihydroxy-8-oxooctadec-12-enoic acid (oxo-DHODE; oxo-Dihydroxy-octadecenoic acid)	C_18_H_32_O_5_	328.44	327		291; 269; 251; 233; 211; 195; 183	279; 258; 247; 236; 217; 195	177; 161	*Bituminaria* [56]; Broccoli [69]; *Phyllostachys nigra* [70]
34	k-6(583); k-632341 (580); k-640 (590)	Unsaturated monocarboxylic acid	Trihydroxyoctadecadienoic acid	C_18_H_32_O_5_	328.44	327		211; 183; 127	183; 167; 149		Potato leaves [27]
35	k-640 (589)	Omega-hydroxy-long-chain fatty acid	Hydroxy docosanoic acid	C_22_H_44_O_3_	356.58	355		309	305; 132		*A. cordifolia* [24]
36	k-1783 (585); k-640 (590)	Steroidal alkaloid	Solanidine	C_27_H_43_NO	397.64		398	185; 272	167		Potato [71,72]
37	k-6(583); k-640 (590)	Long-chain fatty acid	Nonacosanoic acid	C_29_H_58_O_2_	438.77	437		393			*C. edulis* [24]
38	k-1783 (585); k-640 (590)	Steroid	Vebonol	C_30_H_44_O_3_	452.67		453	435; 336; 209	336; 226		*Hylocereus polyrhizus* [73]; *Zostera marina* [74]
39	k-6(583)	Carotenoid	*all-trans*-β-cryptoxanthin caprate		706.2		707	625; 587; 571	527		Sarsaparilla [75]
40	k-640 (590)	Steroidal alkaloid	β-chaconine	C_39_H_63_NO_10_	705.92		706	690			
41	k-6(583)	Carotenoid	(all-E)-violaxanthin myristate		810.1		811	794; 748; 723; 675; 622; 602			Carotenoids [76]
42	k-6(583); k-1783 (585); k-640 (589); k-640 (590); k-642 (582)	Steroidal alkaloid	α-chaconine	C_45_H_73_NO_14_	852.06		852	706	704; 690		*Solanum tuberosum* [72,77,78]
43	k-640 (589); k-642 (582)	Steroidal alkaloid	α-solanine	C_45_H_73_NO_15_	868.96		868	722	560; 398	398; 185	*Solanum tuberosum* [72,77,78]
44	k-6(583); k-640 (589); k-640 (590)	Steroidal alkaloid	Solanidenol chacotriose	C_45_H_73_NO_15_	868.96		868	850; 823; 765; 747; 722; 706	704	677	Potato [77]
45	k-1783 (585)	Steroidal alkaloid	Solanidadiene solatriose	C_45_H_73_NO_15_	868.96		868	706	722; 398; 560		Potato [77]
46	k-6(583); k-640 (590)	Steroidal alkaloid	Solanidenediol chacotriose	C_45_H_73_NO_16_	884.06		884	866; 822; 800; 78; 720; 704	849; 822; 720; 704; 691		Potato [77]
47	k-6(583); k-640 (589); k-640 (590)	Steroidal alkaloid	Leptinine II	C_45_H_73_NO_16_	884.06		884	866; 738; 722	720; 704; 677; 654		*Solanum tuberosum* [77]
48	k-6(583); k-632341 (579); k-632341 (580); k-640 (589); k-640 (590); k-642 (582)	Sapogenin	3-Rhamnose-galactose-glucuronic acid-soyasapogenol B	C_48_H_78_O_18_	943.12	941		615; 733; 795; 923	571		*Bituminaria bituminosa* [56]; *Medicago truncatula* [79]
49	k-6(583); k-640 (589) k-640 (590); k-642 (582)	Sapogenin	6-deoxyhexose-hexoside-uronic acid–soyasapogenol A	C_48_H_78_O_19_	959.12	957		525; 733; 939	457		*Bituminaria* [56]; *Medicago truncatula* [79]

**Table 3 plants-11-02147-t003:** Distribution of bioactive substances in accessions of *V. unguiculata*.

No	Class of Compounds	Identified Compounds	Formula	VIR Catalogue Number				
				k-642	k-632341	k-640	k-1783	k-6
	**Polyphenols**							
1	Flavonol	Dihydrokaempferol (Aromadendrin; Katuranin)	C_15_H_12_O_6_					
2	Flavonol	Quercetin	C_15_H_10_O_7_					
3	Flavonol	Dihydroquercetin (Taxifolin; Taxifoliol)	C_15_H_12_O_7_					
4	Flavonol	Myricetin	C_15_H_10_O_8_					
5	Flavonol	Quercetin 3-*O*-glucoside (Isoquercitrin; Hirsutrin)	C_21_H_20_O_12_					
6	Flavan-3-ol	Epiafzelechin ((epi)Afzelechin)	C_15_H_14_O_5_					
7	Flavan-3-ol	Catechin (D-Catechol)	C_15_H_14_O_6_					
8	Flavan-3-ol	(Epi)afzelechin-4′-*O*-glucoside	C_21_H_24_O_10_					
9	Flavan-3-ol	(Epi)afzelechin-3-*O*-glucoside	C_21_H_24_O_10_					
10	Flavan-3-ol	Chinchonain Ia	C_24_H_20_O_9_					
11	Flavan-3-ol	(epi)Catechin *O*-hexoside	C_21_H_24_O_11_					
12	Flavone	Acacetin (Linarigenin; Buddleoflavonol)	C_16_H_12_O_5_					
13	Tetrahydroxyflavan	Luteoliflavan-eriodictyol-O-hexoside	C_36_H_34_O_16_					
14	Anthocyanidin	Delphinidin 3-*O*-glucoside	C_21_H_21_O_12+_					
15	Anthocyanidin	Delphinidin-3,5-*O*-diglucoside	C_27_H_30_O_17_					
16	Lignan	Dimethylmatairesinol (Arctigenin Methyl Ether)	C_22_H_26_O_6_					
17	Lignan	Medioresinol	C_21_H_24_O_7_					
18	Lignan	Syringaresinol	C_22_H_26_O_8_					
19	Hydroxybenzoic acid (Phenolic acid)	Protocatechuic acid	C_7_H_6_O_4_					
20	Hydroxybenzoic acid (Phenolic acid)	Salvianolic acid D	C_20_H_18_O_10_					
21	Polyphenolic acid	Coumaroyl quinic acid methyl ester	C_17_H_20_O_8_					
22	Derivative of hydroxycinnamic acid	Ferulic acid-*O*-hexoside	C_16_H_20_O_9_					
23	Phenolic acid	*Trans*-salvianolic acid J	C_27_H_22_O_12_					
	**Others**							
24	Non-proteinogenic L-α-amino acid	L-Pyroglutamic acid (Pidolic acid; 5-Oxo-L-Proline)	C_5_H_7_NO_3_					
25	Aminobenzoic acid	4-Aminobenzoic acid (*p*-aminobenzoic acid)	C_7_H_7_NO_2_					
26	Monocarboxylic acid	Dihydroferulic acid	C_10_H_12_O_4_					
27	Carboxylic acid	Indole-3-carboxylic acid	C_10_H_9_NO_2_					
28	Amino acid	L-Tryptophan (Tryptophan; (S)-Tryptophan)	C_11_H_12_N_2_O_2_					
29	Omega-5 fatty acid	Myristoleic acid (Cis-9-Tetradecanoic acid)	C_14_H_26_O_2_					
30	Purine	Adenosine	C_10_H_13_N_5_O_4_					
31	Omega-3 fatty acid	Linoleic acid (Linolic acid; Telfairic acid)	C_18_H_32_O_2_					
32	Hydroperoxy fatty acid	Hydroperoxy-octadecadienoic acid	C_18_H_32_O_4_					
33	Unsaturated monocarboxylic acid	9,10-Dihydroxy-8-oxooctadec-12-enoic acid (oxo-DHODE; oxo-Dihydroxy-octadecenoic acid)	C_18_H_32_O_5_					
34	Unsaturated monocarboxylic acid	Trihydroxyoctadecadienoic acid	C_18_H_32_O_5_					
35	Omega-hydroxy-long-chain fatty acid	Hydroxy docosanoic acid	C_22_H_44_O_3_					
36	Long-chain fatty acid	Nonacosanoic acid	C_29_H_58_O_2_					
37	Steroid	Vebonol	C_30_H_44_O_3_					
38	Carotenoid	*all-trans*-β-cryptoxanthin caprate						
39	Carotenoid	(all-E)-violaxanthin myristate						
40	Steroidal alkaloid	Solanidine	C_27_H_43_NO					
41	Steroidal alkaloid	β-chaconine	C_39_H_63_NO_10_					
42	Steroidal alkaloid	α-chaconine	C_45_H_73_NO_14_					
43	Steroidal alkaloid	α-solanine	C_45_H_73_NO_15_					
44	Steroidal alkaloid	Solanidenol chacotriose	C_45_H_73_NO_15_					
45	Steroidal alkaloid	Solanidadiene solatriose	C_45_H_73_NO_15_					
46	Steroidal alkaloid	Solanidenediol chacotriose	C_45_H_73_NO_16_					
47	Steroidal alkaloid	Leptinine II	C_45_H_73_NO_16_					
48	Sapogenin	3-Rhamnose-galactose-glucuronic acid-soyasapogenol B	C_48_H_78_O_18_					
49	Sapogenin	6-deoxyhexose-hexoside-uronic acid–soyasapogenol A	C_48_H_78_O_19_					

## Data Availability

The data presented in the current study are available in the article.

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
