# Peer review of "Identification and Spatial Distribution of Bioactive Compounds in Seeds Vigna unguiculata (L.) Walp. by Laser Microscopy and Tandem Mass Spectrometry"

_plants, 2022, doi:10.3390/plants11162147_

Round 1
Reviewer 1 Report
1. The manuscript on the identification and spatial distribution of bioactive compounds in V. unguiculata seeds by using microscopy and tandem mass spectrometry is an interesting article.
2. The title contains typographical errors.
3. First sentence of the introduction section is too long and complex. Make it simple and break it into two or three smaller meaningful sentences.
4. Authors should be consistent in the use of nomenclature. They are using both Vigna and cowpea alternatively, which should be avoided. Just use one terminology and be consistent.
5. Authors should arrange the introduction section in a way that makes it coherent. For example, they should start with the species introduction, uses, and cultivation. Then they can talk about its known metabolome composition, especially the seed. Then explain how it is important to know the seed composition for breeding, health, and nutritive purposes. Finally, they can talk about different approaches for such analyses, and explain why they choose the two applied methods. Though this information is already available in the section but is not arranged. Authors need to arrange in a proper coherent way.
6. L112-113. Authors should modify the statement. Change to “However, there is limited information available on the presence and localization of bioactive compounds in cowpea seeds”.
7. From L114 and onward. This section seems too distorted and not arranged properly. Please ask help from a native speaker or English language expert to elaborate these sentences.
8. What means by “Vigna unguiculata rom”.
9. L138. Replace “created” with a more suitable breeding terminology
10. Figure 1. Arrange the figure in a proper way by resizing the panels. Also, since the authors have explained the seed colors in the MM section, therefore, it is recommended to add seed figure panels in Figure 1.
11. L251-55. Cite a relevant article for a Glycine-related statement.
12. What do the different colored texts in Table 1 represent?
13. What do the colored bars in the Table 2 indicate? Is the size of the color bars representing something? I suggest authors prepare a version of the table that is more understandable and self-explanatory.
14. I suggest for explaining the presence/absence of compounds in a specific cultivar can be better explained by a Venn diagram. It would show authors and readers how many metabolites are common between the accessions and how many are unique.
15. The presence/absence of a metabolite can be stage dependent as well as can vary based on the growing conditions. Authors should add statements to tell the readers that all seeds were mature and kept in similar storage conditions to avoid confusion.
16. L327. Replace biologically active substances with “bioactive compounds”.
17. Figure 6. Quality is too bad. Please improve figure quality.
18. Figures from 7 and onward, the figure panels should be adjusted to equal size and rearranged for better representation.
Author Response
Dear Colleagues!
Dear Reviewer!
We sincerely thank you for the detailed work on the text and the identification of errors and various inaccuracies. We have worked on the test and tried to fix everything as fully as possible.
The title contains typographical errors.
Thanks for finding the error. Typographical error removed from article title
3. First sentence of the introduction section is too long and complex. Make it simple and break it into two or three smaller meaningful sentences.
Thanks a lot for the fix. Phrase completely changed.
Vigna unguiculata (L.) Walp. is an important component of farming systems in many parts of the world. It is mainly grown on the continents of Africa, Asia and South America. In recent years, information has appeared about the successful experience of its cultivation in the southern regions of Russia and Russian Far East [1, 2]. V. unguiculata is a multipurpose vegetable crop and it is valued for its drought and heat tolerance. It is grown mainly for its seeds (cvg. unguiculata = ssp. unguiculata) or green vegetable pods (cvg. sesquipedalis = ssp. sesquipedalis). Food products prepared from Vigna are a source of many nutrients: proteins, amino acids, carbohydrates, minerals, fiber, vitamins and other bioactive compounds [3, 4, 5, 6, 7].
4. Authors should be consistent in the use of nomenclature. They are using both Vigna and cowpea alternatively, which should be avoided. Just use one terminology and be consistent.
Yes, indeed, we were inattentive and very inconsistent in the use of nomenclature. Now throughout the text we use the same terminology.
5. Authors should arrange the introduction section in a way that makes it coherent. For example, they should start with the species introduction, uses, and cultivation. Then they can talk about its known metabolome composition, especially the seed. Then explain how it is important to know the seed composition for breeding, health, and nutritive purposes. Finally, they can talk about different approaches for such analyses, and explain why they choose the two applied methods. Though this information is already available in the section but is not arranged. Authors need to arrange in a proper coherent way.
Thanks a lot for the remarks. We completely rewrote the Introduction section and reorganized it.
6. L112-113. Authors should modify the statement. Change to “However, there is limited information available on the presence and localization of bioactive compounds in cowpea seeds”.
We changed the phrase:
We have not found the use of this method for studying the localization of bioactive compounds in V. unguiculata seeds. However, there is limited information available on the presence and localization of bioactive compounds in V. unguiculata seeds.
7. From L114 and onward. This section seems too distorted and not arranged properly. Please ask help from a native speaker or English language expert to elaborate these sentences.
Thank you very much for your remark. Yes, we really are not native English speakers and sometimes we simply do not see incorrectly composed phrases or very “heavy” phrases. We tried to reformat the section.
8. What means by “Vigna unguiculata rom”.
Thanks a lot for the fixes. The phrase has been changed.
The object of the research was the seeds of V. unguiculata from the group of varieties (cultivar groups) sesquipedalis and unguiculata harvested in 2020, grown at the Far East Experiment Station Branch of the Federal Research Center the N.I. Vavilov All-Russian Institute of Plant Genetic Resources (Table 1, Fig. 1).
9. L138. Replace “created” with a more suitable breeding terminology
Thanks a lot for the fixes. The phrase has been changed.
Landraces were collected by N.I. Vavilov during the 1929 expedition to China (k-640, k-642) and obtained from the extract in 1921 from the USA (k-6) and in 1985 from Germany (k-1783), modern cultivar “Lyanchihe” (k-632341) was developed in Russia in the Primorsky Territory as a result of selection from samples of Chinese origin.
10. Figure 1. Arrange the figure in a proper way by resizing the panels. Also, since the authors have explained the seed colors in the MM section, therefore, it is recommended to add seed figure panels in Figure 1.
Thank you very much for this comment. The location of the figure 1. and their correspondence have been changed.
11. L251-55. Cite a relevant article for a Glycine-related statement.
In this case, we are talking about the fact that sapogenins were first identified by us in cowpea. That well-known statement that sapogenins A and B are identified in soybeans by many authors with various research methods is a well-known fact.
12. What do the different colored texts in Table 1 represent?
Thank you for pointing out the error. The text color is made all the same black.
13. What do the colored bars in the Table 2 indicate? Is the size of the color bars representing something? I suggest authors prepare a version of the table that is more understandable and self-explanatory.
The colored columns in the table give a good comparison of similarities and differences in the main polyphenolic groups and groups of other bioactive compounds when comparing different samples of Vigna grown in the experimental fields.
14. I suggest for explaining the presence/absence of compounds in a specific cultivar can be better explained by a Venn diagram. It would show authors and readers how many metabolites are common between the accessions and how many are unique.
Additionally, we presented a Venn diagram for better clarity (Figure 6.).
15. The presence/absence of a metabolite can be stage dependent as well as can vary based on the growing conditions. Authors should add statements to tell the readers that all seeds were mature and kept in similar storage conditions to avoid confusion.
Thank you for pointing out the mistake. Changes have been made.
16. L327. Replace biologically active substances with “bioactive compounds”.
Thank you for pointing out the mistake. Changes have been made.
17. Figure 6. Quality is too bad. Please improve figure quality.
We have slightly altered this drawing. Further improvement is not possible.
18. Figures from 7 and onward, the figure panels should be adjusted to equal size and rearranged for better representation.
Ok, figure panels were adjusted to equal size and rearranged for better representation.
Reviewer 2 Report
The presented manuscript presents a study of the identification and spatial distribution of bioactive compounds in seeds Vigna unguiculata using tandem mass spectrometry and laser microscopy. As a result, new and known compounds were idenetifified in the plant seeds, and their spatial distribution was described. Moreover, the metabolon of different plant varieties was compared. The analysis and conclusion were conducted with concern and precision. The described issue seems very significant in the context of the search for natural-origin compounds in food and foodstuffs. I read the work thoroughly, and I cannot point out any substantial allegations against the research design.
However, some remarks concerning the structure of the manuscript should be made:
- There is an error in the title.
- Line 34 Should be: Steroidal alkaloids have been identified in cowpea seeds for the first time.
- Line 47 “Currently, in the world, …” Phrase “In the world” is too general. This should be changed, as not giving specific information, for example, “in the scientific world”. There are still many places on the globe where the conduction of plant analysis or science activity is impossible due to poverty or the state of the war.
- Lines 162-163 The sentence “HPLC was performed using Shimadzu LC-20 Prominence HPLC 163 (Shimadzu, Japan) was used ..” is incorrect.
- Line 185 and in the entire manuscript: “m/z” should be written in italic.
- Line 244-246 By comparing with what? It is written in the Material and Method section but should also be described in this part for the reader's convenience. Especially since the following sentence in line 245 contains such details.
- Comparing the MS/MS spectrum and data from the articles or the library always gives us tentative structure elucidation. It can not be excluded that the structure of the compounds differs, for example, by the type o the sugar or the chemical bonds conformation, even if the fragmentation pattern is identical and repetitive, and the identification is highly probable. Therefore, the phrase “tentative identification” should be incorporated into the text as the authors prefer.
- Line 261 Alpha-chaconine - alpha should be written with the usage of the appropriate symbol (the same in the entire text and for other compounds like solanine)
- Fig 6 - the figure of better quality should be provided.
- Line 571 - Flavan-3-ol, Anthocyanidin, Lignan, Phenolic acid, the compounds' names should be written in small letters.
Author Response
Dear Colleagues!
Dear Reviewer!
We sincerely thank you for the detailed work on the text and the identification of errors and various inaccuracies. We have worked on the test and tried to fix everything as fully as possible.
However, some remarks concerning the structure of the manuscript should be made:
1. There is an error in the title.
Thanks for finding the error.
Typographical error removed from article title.
2. Line 34 Should be: Steroidal alkaloids have been identified in cowpea seeds for the first time.
Thank you very much for finding the error. The phrase has been completely changed.
Steroidal alkaloids have been identified in V. unguiculata seeds for the first time.
3. Line 47 “Currently, in the world, …” Phrase “In the world” is too general. This should be changed, as not giving specific information, for example, “in the scientific world”. There are still many places on the globe where the conduction of plant analysis or science activity is impossible due to poverty or the state of the war.
Thank you very much for your remark. The paragraph has been completely changed.
Vigna unguiculata (L.) Walp. is an important component of farming systems in many parts of the world. It is mainly grown on the continents of Africa, Asia and South America. In recent years, information has appeared about the successful experience of its cultivation in the southern regions of Russia and Russian Far East [1, 2].
4. Lines 162-163 The sentence “HPLC was performed using Shimadzu LC-20 Prominence HPLC 163 (Shimadzu, Japan) was used ..” is incorrect.
Thank you very much for your remark. The phrase has been completely changed.
HPLC was performed using Shimadzu LC-20 Prominence HPLC (Shimadzu, Japan), equipped with an UV-sensor and a Shodex ODP-40 4E reverse phase column to separate of multicomponent mixtures.
5. Line 185 and in the entire manuscript: “m/z” should be written in italic.
Thank you very much for your remark. All m/z corrected.
6. Line 244-246 By comparing with what? It is written in the Material and Method section but should also be described in this part for the reader's convenience. Especially since the following sentence in line 245 contains such details.
Thank you very much for the clarification. All necessary changes have been made to the main text of the article.
7. Comparing the MS/MS spectrum and data from the articles or the library always gives us tentative structure elucidation. It can not be excluded that the structure of the compounds differs, for example, by the type o the sugar or the chemical bonds conformation, even if the fragmentation pattern is identical and repetitive, and the identification is highly probable. Therefore, the phrase “tentative identification” should be incorporated into the text as the authors prefer.
Thank you very much for your remark. The phrase has been completely changed.
Tentative identification showed the presence of 49 bioactive compounds detected by mass spectrometric analysis in V. unguiculata extracts. 49 target analytes were successfully identified by comparing fragmentation patterns and retention times, most of which were polyphenols.
8. Line 261 Alpha-chaconine - alpha should be written with the usage of the appropriate symbol (the same in the entire text and for other compounds like solanine)
Thank you very much for pointing out the errors. All necessary changes have been made to the text.
9. Fig 6 - the figure of better quality should be provided.
Additionally, we presented a Venn diagram for better clarity (Figure 6.).
10. Line 571 - Flavan-3-ol, Anthocyanidin, Lignan, Phenolic acid, the compounds' names should be written in small letters.
Thank you very much for pointing out the errors. All necessary changes have been made to the text.